

# Comparative transcriptome analysis revealed genes involved in the fruiting body development of *Ophiocordyceps sinensis*

Xinxin Tong[1,*], Han Zhang[1,*], Fang Wang[1], Zhengyao Xue[2], Jing Cao[1], Cheng Peng[1] and Jinlin Guo[1]

[1] Key Laboratory of Standardization of Chinese Medicine, Ministry of Education; Key Laboratory of Systematic Research, Development and Utilization of Chinese Medicine Resources in Sichuan Province-Key Laboratory Breeding Base founded by Sichuan Province, Chengdu University of Traditional Chinese Medicine, Chengdu, Sichuan, China

[2] Department of Food Science and Technology, University of California, Davis, CA, United States of America

[*] These authors contributed equally to this work.

Corresponding author
Jinlin Guo, guo596@cdutcm.edu.cn

## ABSTRACT

*Ophiocordyceps sinensis* is a highly valued fungus that has been used as traditional Asian medicine. This fungus is one of the most important sources of income for the nomadic populations of the Tibetan Plateau. With global warming and excessive collection, the wild *O. sinensis* resources declined dramatically. The cultivation of *O. sinensis* hasn't been fully operational due to the unclear genetic basis of the fruiting body development. Here, our study conducted pairwise comparisons between transcriptomes acquired from different growth stages of *O. sinensis* including asexual mycelium (CM), developing fruiting body (DF) and mature fruiting body (FB). All RNA-Seq reads were aligned to the genome of *O. sinensis* CO18 prior to comparative analyses. Cluster analysis showed that the expression profiles of FB and DF were highly similar compared to CM. Alternative splicing analysis (AS) revealed that the stage-specific splicing genes may have important functions in the development of fruiting body. Functional enrichment analyses showed that differentially expressed genes (DEGs) were enriched in protein synthesis and baseline metabolism during fruiting body development, indicating that more protein and energy might be required for fruiting body development. In addition, some fruiting body development-associated genes impacted by ecological factors were up-regulated in FB samples, such as the nucleoside diphosphate kinase gene (*ndk*), β subunit of the fatty acid synthase gene (*cel-2*) and the superoxide dismutase gene (*sod*). Moreover, the expression levels of several cytoskeletons genes were significantly altered during all these growth stages, suggesting that these genes play crucial roles in both vegetative growth and the fruiting body development. Quantitative PCR (qPCR) was used to validate the gene expression profile and the results supported the accuracy of the RNA-Seq and DEGs analysis. Our study offers a novel perspective to understand the underlying growth stage-specific molecular differences and the biology of *O. sinensis* fruiting body development.

## INTRODUCTION

*Ophiocordyceps sinensis* (Berk.), belonging to Ascomycetes, is endemic to the Tibetan Plateau and the Himalayas with high latitudes from 3,000 to 5,000 m (*Lo et al., 2015*). The Chinese name for *O. sinensis* is "Dong Chong Xia Cao", meaning "winter worm, summer plant". The name derived from its unusual lifestyle: entering and growing inside the ghost moth caterpillars (*Thitarodes* spp.), developing the fruiting body from the caterpillar shell, and forming a parasitic fungus-insect complex that comprises a mature fruiting body with perithecium (*Lo et al., 2015*). A full life cycle of *O. sinensis* is typically 3–5 years when growing in the wild (*Qin et al., 2018*). At present, the molecular basis of fruiting body development hasn't been fully understood.

O. sinensis has been widely used as a traditional Asian medicine to treat asthma, bronchitis, lung inflammation, nocturnal emissions, night sweats, and other diseases for over 2000 years (*Xu et al., 2016*). This medicinal fungus is also one of the most important sources of income for the native tribes of the Tibetan Plateau (*Pöggeler, Nowrousian & Kück, 2006*; *Guo, Liu & Kanari, 2012*; *Xia et al., 2017*). Due to the high medical value and huge market demand, the price of mature fruiting body was as high as approximately USD 60,000 per kg for the top grade in 2015 (*Qin et al., 2018*). The natural populations of *O. sinensis* were overexploited, inevitably impairing the sustainability of the regional economy of Tibet and endangering this species (*Qin et al., 2018*). Even though under the optimized artificial cultivation, the fruiting body takes 1–2 years to mature (*Qin et al., 2018*). To date, the large-scale cultivation of *O. sinensis* has not been realized. A better understanding the biology of the fruiting body development is essential for scaling up the cultivation of *O. sinensis* to meet the medicinal demand and reduce the exploitation of wild *O. sinensis* populations.

Fruiting body development of *O. sinensis* requires special environmental factors and has been demonstrated to be initiated by the ecological factors specific to the alpine ecosystem, such as high latitude, low temperature and intensive solar ultraviolet (UV) radiation (*Xia et al., 2017*). The influence of environmental factors on fruiting body development has been investigated in some Ascomycetes fungi, including *Neurospora crassa* and *Pestalotiopsis microspore*, etc. (*Oda & Hasunuma, 1997*; *Lee et al., 2004*; *Pöggeler, Nowrousian & Kück, 2006*). Previous studies showed that both orientation and the position of the neck on the perithecium were light-dependent in *N. crassa* (*Oda & Hasunuma, 1997*). Light controls the balance of asexual versus sexual reproduction of *Aspergillus nidulans* (*Busch et al., 2010*). Moreover, several genes involved in the perception of these environmental factors have been identified, such as nucleoside diphosphate kinase gene and a subunit of COP9 signalosome gene (*Oda & Hasunuma, 1997*; *Busch et al., 2010*). These results provide good references for researches in *O. sinensis*.

With the development of high throughput sequencing technology, the genomes and transcriptomes of *O. sinensis* and related species have been analyzed (*Zhong et al., 2016*; *Zhong et al., 2018*). Recent research detected the signals of the positive selection for genes involved in peroxidase and hypoxia to enable the highland adaptation by sequencing 31 whole genomes of *O. sinensis* from different distributions (*Xia et al., 2017*).

*Xiang et al. (2014)* constructed an *O. sinensis* EST database by using Roche/454 GS-FLX (Titanium) pyrosequencing technology, four mating type genes and 121 genes putatively associated with fruiting body development were identified. Moreover, the transcripts of three developmental stages (mycelia, sclerotium and fruiting body) were sequenced and compared, however, both sclerotium and fruiting body were the worm—part and grass—part of *O. sinensis* respectively, which couldn't represent the different growth stages of *O. sinensis* (*Zhong et al., 2018*). Transcriptome analysis of *O. sinensis* before and after infection of *Thitarodes* offered insights into the infection biology by which the fungus enters into the host partner (*Zhong et al., 2016*).

Built on prior results, in this study, the samples of *O. sinensis* were all harvested from artificial cultivation in our lab, which enabled a convenient sampling of asexual mycelium, the developing stroma and mature fruiting body. With the state-of-art Illumina HiSeq 2000 technology, transcriptome profiles of the three growth stages were sequenced in two biological replicates and compared between different growth stages. Our study revealed significant differences in expression profiles of genes involved in protein biosynthesis and baseline metabolism in these growth stages. Moreover, some fruiting body development associated genes influenced by environmental factors were identified. For the first time, the expression levels of cell cytoskeleton genes were altered throughout the growth stages in this study, indicating that cytoskeletons might be critical for fruiting body formation. Besides, stage-specific splicing genes might play an important roles in fruiting body development. Together, our study provided novel insights into the genetic basis of fruiting body development, which would facilitates the large-scale artificial cultivation of *O. sinensis*.

## MATERIALS AND METHODS

### Specimen collection, RNA extraction and sequencing

Three developmental stages of *O. sinensis* were artificially cultivated and collected in our lab. *O. sinensis* strain was deposited in the China General Microbiological Culture Collection center, accession number CGMCC 3.14243, and cultured on potato dextrose agar (PDA) plate. The asexual mycelium (CM) from PDA plate was harvested for RNA isolation. The samples of stroma without asci was designated as the developing fruiting body (DF). The samples of fruiting body with asci and ascospores were designated as mature fruiting body (FB) and harvested for RNA isolation. All fresh specimen collected were stored in RNAlater® (Ambion, Austin, Texas, USA) according to the manufacturer's instruction until ready for RNA extraction. Total RNA was isolated by using TRIzol (Invitrogen, USA) according to the manufacturer's instruction. Genomic DNA was digested by DNase I (Fermentas, USA). Purified RNA was electrophoresed in a 1% agarose gel, and the purity and the quality of RNA were assessed by OD260 and OD230. At least 20 µg total RNA was then submitted to Biomarker Technology Co., Ltd. (Beijing, China) for quality control using an Agilent Bioanalyzer 2100 (Agilent Technologies, CA, USA).

RNA purity, concentration and RNA integrity number (RIN) were measured by Agilent 2100 Bioanalyzeror SMA3000. The mRNA was enriched from total RNA using poly

(T)+ oligo attached magnetic beads, eluted with Tris–HCl buffer and fragmented in fragmentation buffer using an RNA fragmentation kit (Ambion, Austin, TX, USA). First-strand cDNAs were reverse transcribed with reverse transcriptase and random hexamer-primers, and then second-strand cDNAs were synthesized by DNA Polymerase I.A single 'A' base was added to the ends of these dscDNA, and then the cDNA was ligated with Illumina sequencing adaptors and separated in an agarose gel. The 200 bp cDNA fragments were then purified from the gel using Gel Extraction Kit (Axygen, Biosciences, Central Avenue Union City, CA, USA) for further library preparation using NEB Next Ultra RNA Library Prep Kit for Illumina (NEB, USA) according to the manufacture's instruction. The fragmented cDNA libraries of each sample were then submitted to an Illumina HiSeq 2000 platform at Biomarker Technology Co., Ltd. (Beijing, China) for transcriptome sequencing with the 125 bp paired-end (PE) chemistry. Adapter sequence and low quality bases were removed from the raw reads.

### Read mapping, annotation and differential expression analysis

*O. sinensis* genome was downloaded from the NCBI database (https://www.ncbi.nlm.nih.gov/assembly/GCA_000448365.1/). Trimmed paired-end reads of FB, DF and CM were aligned to the reference genome by the fast, splice-aware HISAT2 aligner (version 2.0.4) (*Kim, Langmead & Salzberg, 2015*). Reads that have been aligned to the genome from each sample were then assembled into transcripts by StringTie (version 1.3.3b) using default parameters (version 2.0.6) (*Langmead, 2010*; *Pertea et al., 2015*). The mapping results are shown in Table 1. Coding sequences were predicted by Transdecoder (version 2.0.1). Functional annotation of assembled sequences was retrieved from the databases of Kyoto Encyclopedia of Genes and Genomes (KEGG, http://www.kegg.jp/) (*Kanehisa & Goto, 2000*), Cluster of Orthologous Groups (COG, http://www.ncbi.nlm.nih.gov/COG/) (*Ashburner et al., 2000*; *Tatusov et al., 2003*), NCBI's non-redundant protein (NR, http://www.ncbi.nlm.nih.gov/sites/entrez?db=protein) and SwissProt (http://www.expasy.org). To globally characterize the expression patterns of diverse RNA-Seq samples, paired-end reads were aligned back to the assembled transcripts using Bowtie 2.0 as the aligner (*Langmead & Salzberg, 2012*). Gene expression patterns were quantified using STAR-RSEM algorithm (version 4.1) based on the read numbers that were mapped to each gene. The mapped read numbers of each assembled transcript were calculated, and were normalized by RESM-based algorithm to get FPKM values for each RNA-Seq sample using perl scripts in the Trinity package (*Pertea et al., 2016*). RSEM results of each replicate of the sample were merged as one matrix for downstream analyses. The new assembled transcripts were deposited into Figshare (DOI: 10.6084/m9.figshare.11086478).

Using the edgeR package (Empirical analysis of Digita Gene Expression in R, http://www.bioconductor.org/packages/elease/bioc/tml/edgeR.html), differentially expressed genes (DEGs) were identified with a threshold of $|\log_2 FC(\log_2 \text{fold-change})| \geq 2$ and false discovered rate (FDR) <0.001 as statistically significant. With the TopGO package (http://www.bioconductor.org/packages/release/bioc/html/topGO.html), enrichment was tested using a hypergeometric test with a threshold value of $p \leq 0.01$ for GO functions and KEGG pathways. The expression patterns of DEGs across the growth stages were further

**Table 1  Mapping results of *O. sinensis* transcriptomes.**

| Sample | Clean reads | Clean bases | GC (%) | % mapped reads to genome |
|---|---|---|---|---|
| CM-1 | 35,128,504 | 4.43G | 60.40 | 83.41 |
| CM-2 | 36,788,192 | 4.63G | 60.00 | 83.98 |
| DF-1 | 38,627,930 | 4.87G | 60.70 | 82.61 |
| DF-2 | 34,270,616 | 4.32G | 61.00 | 80.77 |
| FB-1 | 32,751,220 | 4.57G | 60.35 | 85.12 |
| FB-2 | 36,296,836 | 4.13G | 59.93 | 83.84 |

clustered using hierarchical clustering implemented in R package. For each cluster, GO term enrichment analysis was conducted using TopGO package with a threshold value of $p \leq 0.01$.

Raw Illumina sequencing results of *O. sinensis* were submitted to NCBI Sequence Read Archive (SRA) with the accession numbers: SRR5282569, SRR5282570, SRR5282574, SRR5282575, SRR5282577 and SRR5282578.

## Alternative splicing prediction

ASprofile (version b-1.0.4, http://ccb.jhu.edu/software/ASprofile/) was used to detect the splice junction sites and provided information about boundaries and combinations of different exons in a transcript, and then total splice junction sites of the same gene were used to distinguish the type of its alternative splicing (AS) event. The ASprofile analysis was deposited into Figshare (DOI: 10.6084/m9.figshare.11086478). Using rMATs (version 4.0.3, http://rnaseq-mats.sourceforge.net/), the number of differential splicing events (DASs) and different splicing genes (DSGs) were compared between different growth stages and FDR < 0.05 as statistically significant. TopGO was used to perform function enrichment analysis of DSGs between different growth stages and $p \leq 0.01$ was used to judge the significant enriched GO terms.

## Quantitative RT-PCR

Ten genes that were randomly selected between the three growth stages according to the transcriptome sequencing analysis were selected and validated by qPCR experiments in triplicate. Primers were designed according to the sequences acquired from Illumina sequencing results with Primer Premier 5.0. The specific primer sequences are listed in Supplemental Information 11. Aliquots of cDNA template submitted for Illumina HiSeq sequencing were used for qPCR analysis.. cDNA amplification was performed in a total of 25 µL reaction system containing 12.5 µL 2 × UltraSYBR qPCR Mix (Cwbiotech, Co., LTD., Beijing, China), 2 µL of diluted cDNA mix, 0.5 µL of each primer (10 mM) .The cycling conditions were as follows: 95 °C for 10 min, 40 cycles of 95 °C for 10 s, 57 °C for 15 s and 72 °C for 20 s. Relative gene expression levels were analyzed by the $2^{-\Delta\Delta Ct}$ method. The histone 2A gene (*MSTRG. 2474*) was used to normalize the expression level. Graphpad Prism program (version 5.0, GraphPad Software, Inc.) was used for statistical analysis.

## RESULTS

### RNA sequencing and mapping

To investigate the profile of gene expression during growth, samples with two biological replicates from the stages of (CM, DF and FB) were submitted for RNA-Seq. After cleaning and quality control, more than 63 million paired-end, clean reads were obtained for each replicate with Illumina HiSeq. Reads of CM, DF and FB were aligned to the *O. sinensis* CO18 genome using HISAT2 (*Kim, Langmead & Salzberg, 2015*), and assembled by StringTie (*Pertea et al., 2015*), which resulted in 11,497 genes, including 7518 protein coding genes and 233 putative novel genes (Supplementary Information 1). The number of aligned reads per gene was normalized by the STAR-RSEM based algorithm (*Li & Dewey, 2011*). For each replicate, over 80.77% reads were successfully mapped (Table 1). In a previous study, Li Xiang et al. constructed an *O. sinensis* EST database using Roche/454 GS-FLX (Titanium) pyrosequencing technology and they obtained a total of 1,743,676 high-quality reads with an average length of 426 bp, and assembled these reads into 34,289 unique sequences with an average length of 612 bp (*Xiang et al., 2014*). BLAT comparison was performed between the 11,497 genes in this study and 34,289 sequences in Li's study (*Xiang et al., 2014*), which revealed that more than 90.00% of the 34,289 sequences can be found in the 11,497 genes with an identity of 90%, while only 78.01% of the 11,497 genes can be found in the 34,289 sequences, indicating that our assembly identified more genes than the previous study.

### Differential gene expression and functional enrichment analysis

DEGs were considered statistically significant if the $|\log_2FC(\log_2 fold\text{-}change)|$ is more than 2 and the FDR is less than 0.001. This threshold resulted in a total of 1153 genes as significant DEGs in DF vs. CM , 503 in FB vs. DF and 1234 in FB vs. CM (Fig. 1A, Supplemental Information 1). There were 19 shared DEGs among the three comparisons of the different growth stages (Fig. 1B, Supplemental Information 2). Among them, one gene (*MSTRG.3745*) encoding one hypothetical protein with the homology of 60s ribosomal protein L3 in *O. sinensis* CO18 was significantly up-regulated in FB compared to that in both CM and DF. However, most of the shared genes encode functionally unknown proteins, which might be related to fruiting body development and required further study to verify their functionalities.

Based on both KEGG and GO databases, we analyzed the enrichment of the DEGs across the three growth stages (Supplemental Information 3). In DF vs. CM comparison, a total of 98 DEGs were mainly enriched in 'MCM complex', 'ATP catabolic process', 'ATP-dependent helicase activity', 'ATP binding', DNA replication initiation', 'replication compartment', and 'small-subunit processome' etc., among which 78 and 20 genes were up-regulated and down-regulated in DF, respectively (Supplemental Informations 3 and 4), suggesting that the increased demand of energy and protein for fruiting body formation compared to the asexual mycelium stage. In FB vs. DF comparison, 161 DEGs were enriched in ' ribosome biogenesis', 'structural constituent of ribosome', 'ribonucleoprotein complex', 'MCM complex', 'translation', etc. were enriched (Fig. 2, Supplemental Information 3), among which 103 and 58 genes up- and down-regulated in FB, respectively (Supplemental

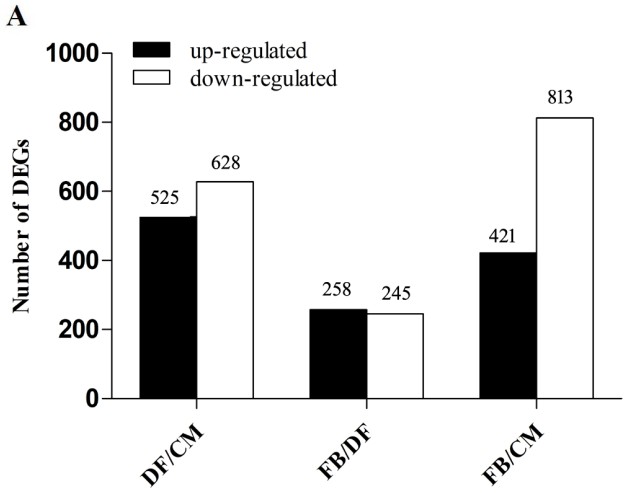

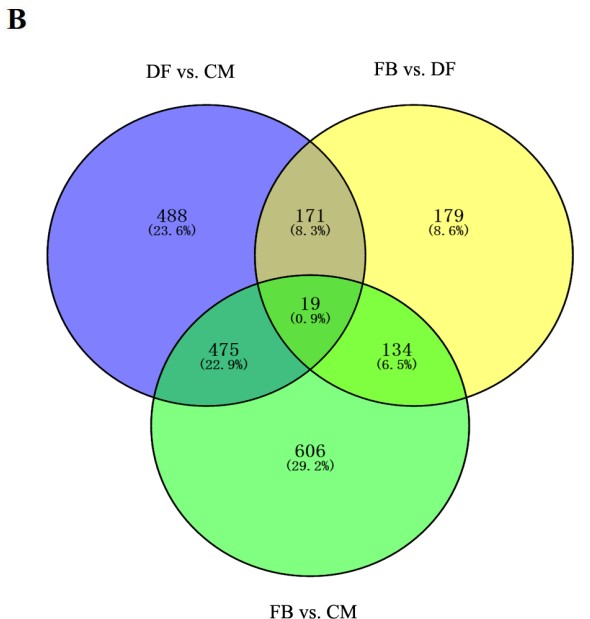

**Figure 1  Analysis of differentially expressed genes (DEGs) between different growth stages.** (A) The number of DEGs is shown on the top of histograms. Statistics of DEGs from *O. sinensis* between different growth stages. (B) Venn diagram of DEGs comparing between different growth stages from *O. sinensis*. FB represents the mature fruiting body. CM represents the asexual mycelium. DF represents the developing fruiting body.

Information 4). Moreover, a total of 23 genes enriched in ribosome biogenesis were all significantly up-regulated in FB compared to DF (Supplemental Information 3). Besides, in FB vs. CM comparison, 304 genes were mainly enriched in 'structure constituent of ribosome', 'ribosome biogenesis', 'translation', 'small ribosomal subunit', 'oxidoreductase activity', 'carbohydrate metabolic process', 'pyridine-containing/organic hydroxy compound biosynthetic process' and 'iron ion binding', etc. (Fig. 2, Supplemental
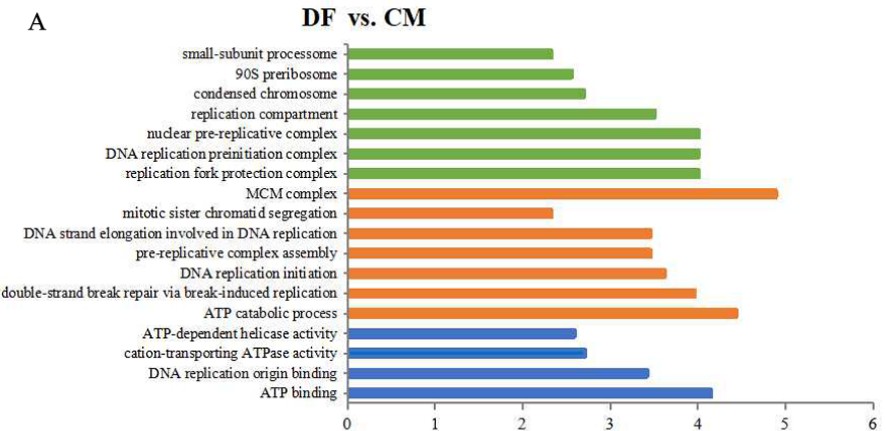

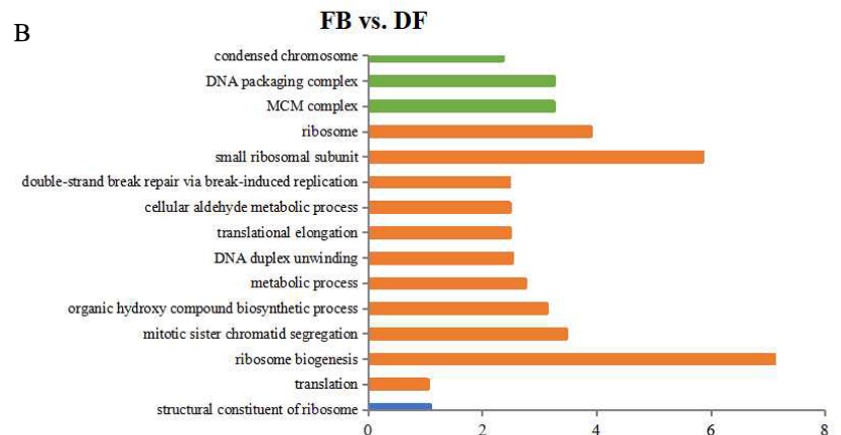

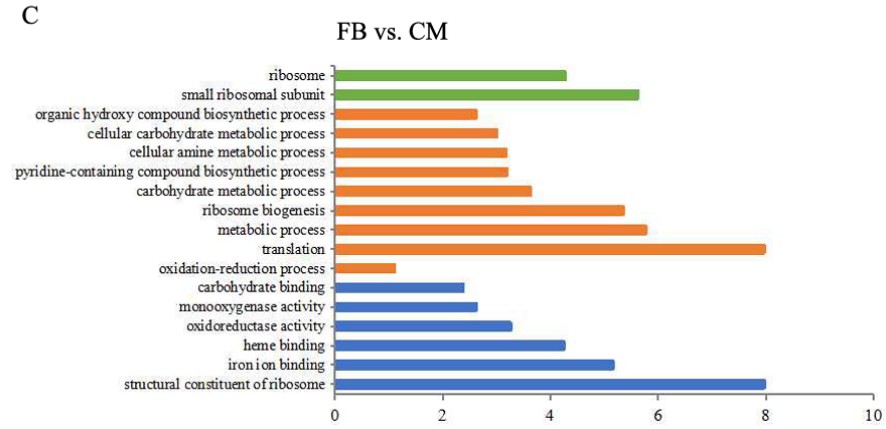

**Figure 2  The most enriched GO functional classification of DEGs between different growth stages.** Only the significant GO terms ($p < 0.005$) were shown. $X$-axes represent the enrichment score ($-Log_{10}$ $P$-value) of top GO terms enriched among DEGs. The green bars represent cellular components; The orange represent biological processes; the blue bars represent molecular functions. FB represents the mature fruiting body. CM represents the asexual mycelium. DF represents the developing fruiting body.

Information 3), in which 132 and 172 genes were up-regulated and down-regulated in FB, respectively (Supplemental Information 4).

## Clustering of gene expression profiles across three growth stages

To verify the correlation between the expression of all the DEGs of developmental stages, principle component analysis (PCA) showed that FB shared more similar expression profiles with DF than CM (Fig. 3A). Furthermore, hierarchical clustering was performed on the basis of the expression level of DEGs across these growth stages, revealing that the three growth stages with two biological replicates could be assigned into two groups with FB and DF as one group and CM as the other (Supplemental Information 5) and overall gene expression pattern was divided into seven gene clusters with visibly different profiles (Figs. 3B–3H). Cluster 1 and cluster 3, with 712 and 72 transcripts, respectively, had a sudden decrease in expression upon transition from CM to DF, and a steady trend from DF to FB, indicating that these genes were up-regulated in CM and might play crucial roles in the growth of mycelium. Cluster 4, with 188 transcripts, had a steady trend from CM to DF, and then a sudden decrease in expression from DF to FB, illustrating that these genes mainly were expressed in CM and DF. Cluster 1, 3 and 4 were mainly enriched in genes involved in 'carbohydrate metabolic process', 'oxidoreductase activity' and 'protein kinase activity', which might contribute to the vegetative growth and physiological metabolism (Supplemental Information 6). Cluster 2, with 139 transcripts, had a sudden decrease in expression upon shifting from CM to DF and a sudden increase from DF to FB. Cluster 6, with 280 transcripts, had a sudden increase in expression from CM to DF, and a sudden decrease from DF to FB, which indicated that these genes have different roles during the three stages. Cluster 5, with 471 transcripts, remained a stead trend from CM to DF, and then a sudden increase in expression upon transition from DF to FB. Cluster 5 mainly consisted of genes involved in 'ribosomal structure constituent/biogenesis', 'translation', 'starch/sucrose metabolic process', 'polysaccharide catabolic process', etc. (Supplemental Information 6). These genes were at the highest expression level in FB compared to DF and CM. Cluster 7, with 280 transcripts, had a sudden increase in expression upon transition from CM to DF, and then remained a steady trend between DF and FB, consisting of genes mainly involved in 'ribosome', 'ATP catabolic process' and 'protein metabolism process' (Supplemental Information 6), such as $Ca^{2+}$ transporting ATPase (*MSTRG.2959*) and ATP-dependent RNA helicase (*MSTRG.10766*).

## Alternative splicing prediction

AS events can increase the complexity and diversity of genes (*Wang et al., 2015*). Some basic AS events, including alternative 3′ splice receptor (A3SS), alternative 5′ splice donor (A5SS), intron retention (IR) and exon skipping (SKIP) (*Wang et al., 2015*), were investigated in this study. Our data identified 2581 AS events in CM, 2635 in DF and 2203 in FB (Fig. 4A). IR is the major class which accounted for over 44.35% of these AS events in *O. sinensis.* When comparing AS events between different growth stages of *O. sinensis*, we found that 946, 652 and 937 DASs related to 783, 566 and 780 DSGs were found to be significantly different in DF vs. CM, in FB vs. DF and in FB vs. CM, respectively (Figs. 4B

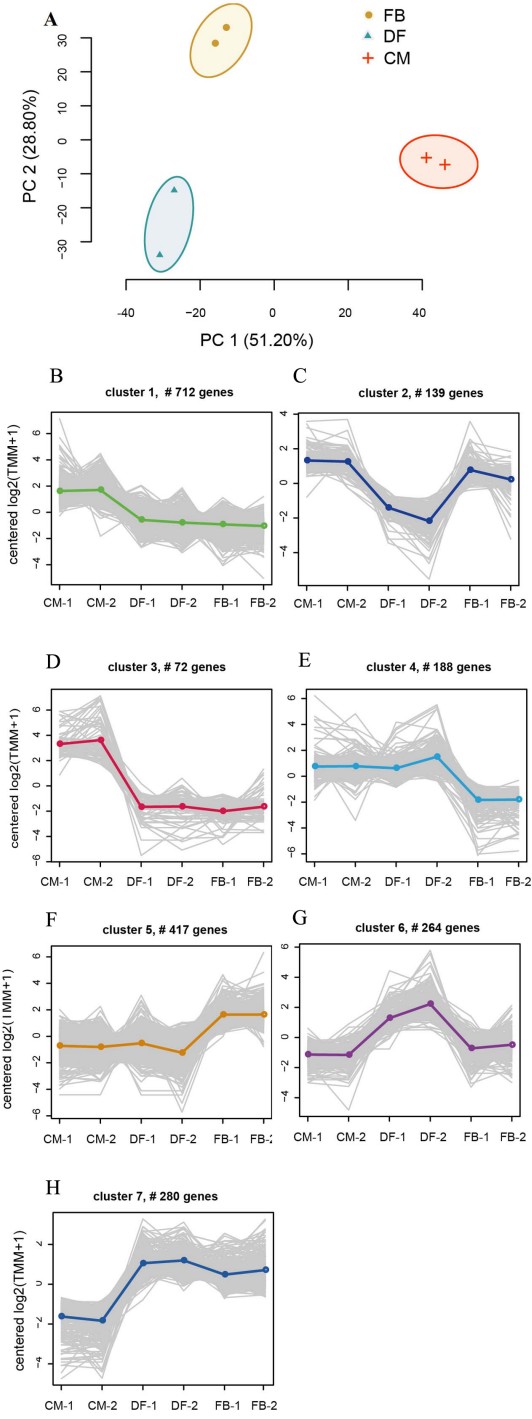

**Figure 3** **Clustering of gene expression profiles across the three growth stages.** (A) Principle component analysis of the RNA-Seq data. (B–H) Six gene clusters with different expression patterns. Overlaying curves of all genes within the cluster were shown. FB represents the mature fruiting body. CM represents the asexual mycelium. DF represents the developing fruiting body.

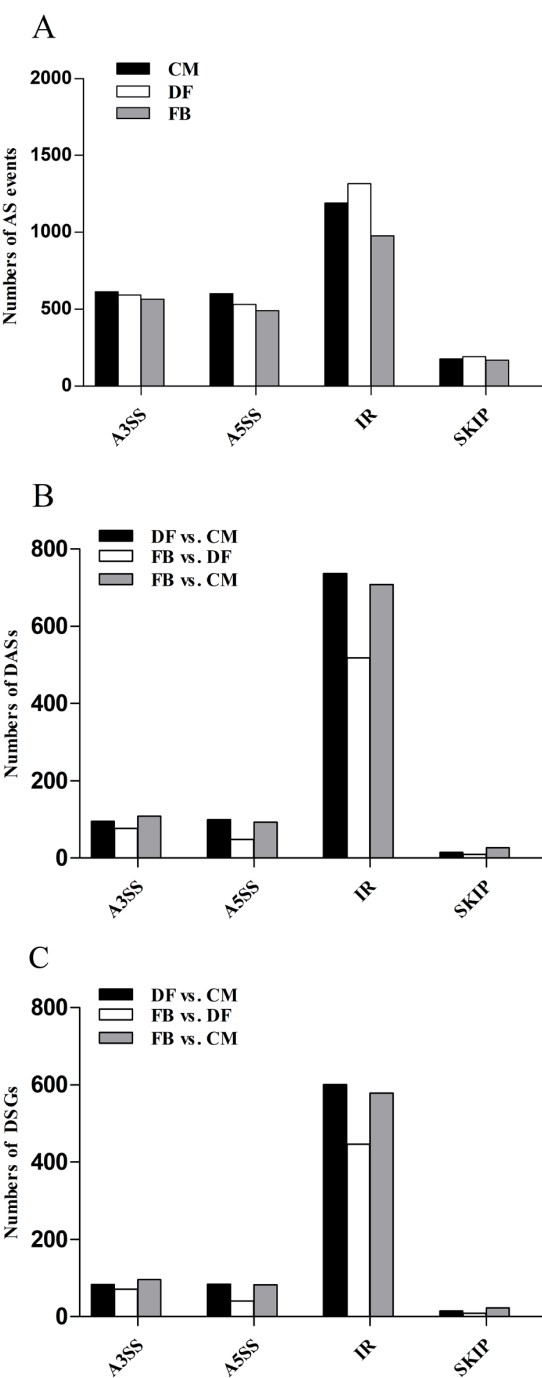

**Figure 4  AS events in the *O. sinensis* transcriptome.** (A) Numbers of alternative splicing (AS) events in different growth stages of *O. sinensis*. (B) Numbers of differential alternative splicing events (DASs) between different stages. (C) Numbers of differential splicing genes (DSGs) between different stages. The *x*-axis represents AS types. FDR < 0.05. FB represents the mature fruiting body with mature. CM represents the asexual mycelium. DF represents the developing fruiting body. IR, intron retention; A3SS, alternative 3′ splice receptor; A5SS, alternative 5′ splice donor; SKIP, exon skipping.

and 4C, Supplemental Information 7). IR-related DSGs is the major class in all types of DSGs (Fig. 4C, Supplemental Information 7). Then the function enrichment analysis of DSGs showed that in DF vs. CM, DSGs were mainly enriched in 'tryptophan/tyrosine/L-phenylalanine biosynthetic process'. In FB vs. DF, DSGs were enriched in 'leucine and valine biosynthetic process' (Supplemental Information 8). In FB vs. CM, DSGs were enriched in 'damaged DNA binding', 'oxidoreductase activity/reduction of molecular oxygen' and ' heme binding' (Supplemental Information 8). Moreover, there are 173 shared DSGs among the three growth stages (Supplemental Information 9), such as a gene encoding Hsp 90 binding chaperon (*mod-E, MSTRG.8149*) and a eukaryotic translation initiation factor gene (*MSTRG.6510*), both of which were significantly up-regulated in FB by 5.24-fold and 5.04-fold compared to CM, respectively.

## Analysis of cell cytoskeleton genes

The transcriptome analysis showed that the expression levels of cell cytoskeleton genes were altered during the three serial growth stages. One actin gene (*MSTRG.3317*, Supplemental Information 1) was down-regulated by 3.43-fold in FB compared to DF, and two tubulin/FtsZ protein genes (*MSTRG.3823* and *MSTRG.6894*, Supplemental Information 1) were up-regulated by 3.04- and 2.90-fold in FB compared to DF, respectively. *TubA* and *tubB* are essential for mitosis, nuclear migration and ascospore formation (*Kirk & Morris, 1991*). One actin related protein gene (*MSTRG. 3090*, Supplemental Information 1) was up-regulated by 3.66-fold in FB compared to DF, while in DF vs. CM, the gene was down-regulated by about 2.86-fold in DF compared to CM.

## Analysis of development-related genes regulated by environmental factors

Fruiting body development is a complex cellular differentiation process, which is regulated by special environmental stimulus. In our study, several putatively development-related genes putatively regulated by light were also identified (Supplemental Information 10). For examples, one nucleoside diphosphate kinase gene (*ndk , MSTRG. 2107*), was up-regulated by 6.23-fold in FB compared to DF. Besides, three of superoxide dismutase genes (*sod, MSTRG. 2218, MSTRG. 5954* and *MSTRG.6951*, Supplemental Information 10) were remarkably up-regulated in FB compared to DF. Our data showed that the expression level of COX gene (*MSTRG.3789*, Supplemental Information 10) increased in FB at 6.44-fold higher expression level than that in DF. Besides, our data showed that the expression level of the NADPH oxidase gene (*MSTRG.7071*, Supplemental Information 10) was significantly higher in FB at 4.06-fold of DF and 4.08-fold of CM.

In addition, low temperature is another physical factor influencing fruiting body development. Several genes regulated by cold were identified in this study, such as β subunit of fatty acid synthase gene (*cel-2, MSTRG.6193*, Supplemental Information 10) and acetyl-CoA desaturase gene (*MSTRG.3201*, Supplemental Information 10). In our study, the expression level of *cel-2* and acyl -CoA desaturase gene were up-regulated in FB by 6.36-fold and 6.42-fold compared to CM, respectively. In FB vs. DF, acyl-CoA desaturase gene was expressed at 6.52-fold higher level in FB than that in DF (Supplemental Information

10). Besides, two genes (*MSTRG.4136, MSTRG.10796,* Supplemental Information 10) encoding low-temperature enzyme were potentially identified to be heat-shock protein (Hsp) 70 chaperone, which was supported by the evidence that, Mod-E, a Hsp 90 homolog, was reported to be involved in the temperature-dependent sexual development in *P. anserina* (*Loubradou, Bégueret & Turcq, 1997*). Moreover, four heat shock protein 101 genes (*MSTRG.11220, MSTRG.10780, MSTRG.10779* and *MSTRG.1122,* Supplemental Information 10) were found to be significantly up-regulated in FB and DF compared to CM.

## Analysis of signal transduction pathway

The fruiting body development was regulated by specific environmental stimuli, and subsequently regulated by signal transduction pathways. In our study, a total of 13 genes encoding MAPKKK, MAPKK and MAPK were identified in the transcriptome during the growth periods. Our data found that four of five MAPKKK encoding genes (*MSTRG.10919, MSTRG.5562, MSTRG. 8422* and *MSTRG.3582,* Supplemental Information 1), were expressed at higher levels at stages of DF and FB compared to CM. *MSTRG.6508* was annotated as MAPK, and *MSTRG.3890* as WD domain containing protein, a MAPK binding protein. Both *MSTRG.3890* and *MSTRG. 6508* were expressed at higher levels in FB compared to CM (Supplemental Information 10). Some serine/threonine-protein kinase genes (e.g., *MSTRG.9127, MSTRG.6087, MSTRG.3872* and *MSTRG.3872*, FDR >0.001, Supplemental Information 1) were at the tendency of higher expression levels in FB and DF compared to CM. Afterwards, these activated MAPK protein would activate the downstream transcript Factors (TFs) to regulate the fruiting body development. Additionally, PKA pathway also leads to ascocarp formation and is associated with the asexual development, hyphal growth, and pathogenicity in several Ascomycetes species (*Pöggeler, Nowrousian & Kück, 2006*). One cAMP-dependent protein kinase encoding gene (*MSTRG.2256,* Supplemental Information 1) is up-regulated by 2.34-fold in CM compared to DF. Taken together the fruiting body development in *O. sinensis* might be more dependent on the MAPK pathway than the cAMP—dependent PKA pathway.

## Validation of transcriptome data by quantitative RT-PCR analysis

To confirm the reliability of RNA-Seq analysis, the expression level of the genes were randomly selected and analyzed by qPCR during three development stages. In FB vs. DF, except for two genes (*MSTRG.5396* and *MSTRG.5638*), the expression changes of eight genes detected by qPCR is similar to the direction of fold change acquired by the Illumina sequencing results, although the fold changes detected by qPCR wasn't completely consistent with those determined by RNAseq analysis, including six genes up-regulated in FB and two genes up-regulated in DF (Fig. 5A). In DF vs. CM comparison, except for two genes (*MSTRG.5638* and *MSTRG.5297,* Supplemental Information 1) , the qPCR results of seven genes were similar to the results of transcriptome analysis, including four genes up-regulated in CM, two gene up-regulated in DF and one genes similarly expressed between the two stages (Fig. 5B). These results showed that our transcriptome data are reliable.

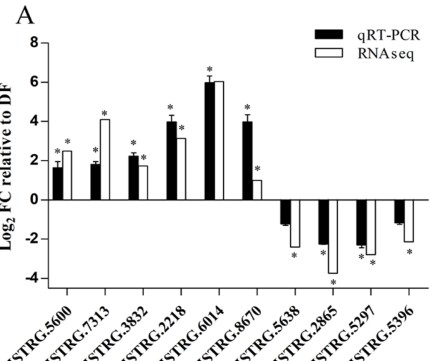
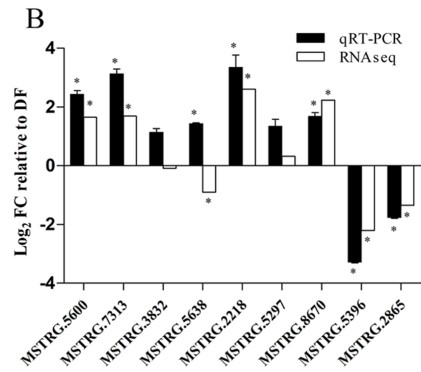

**Figure 5  qPCR validation of the expressed genes in Illumina sequencing.** (A) Bars represent the fold change in expression of each candidate gene identified in FB relative to DF. (B) Bars represent the fold change in expression of each candidate gene identified in CM relative to DF Black bars represent qRT-PCR result ($2^{-\Delta\Delta Ct}$). Error bars indicate the standard error. White bars represent the RNA-seq results (log₂ fold change). Histone H2A gene (MSTRG. 2474) was the internal reference. FB represents the mature fruiting body. CM represents the asexual mycelium. DF represents the developing fruiting body. * represents significant difference in gene expression between FB/CM and DF, respectively, with qRT-PCR measured by paired $t$-test at $p < 0.05$ and RNAseq measured by edgeR at FDR $< 0.05$.

## DISCUSSION

Here, we report the transcriptomes of three different growth stages in *O. sinensis*. The DEGs analysis revealed that the expression patterns of the fruiting body developmental stages (FB and DF) were more similar compared to the vegetative CM stage. Moreover, the percentage of up-regulated DEGs enriched in ATP process, MCM complex and DNA replication is much higher than that of the down-regulated genes in DF compared to CM, indicating that the increased demand of energy and protein for fruiting body formation. The percentages of DEGs enriched in ribosomal biogenesis and translation were much higher than that of the down-regulated in FB compared to DF. Previous studies indicated that growth media lacking tryptophan or histidine would lead to loss of cleistothecia formation in *Aspergillus nidulans* (*Eckert et al., 1999*; *Eckert et al., 2000*; *Busch et al., 2001*). Moreover, deletion of the tryptophan synthase-encoding gene *trpB*, or the histidine biosynthesis gene *hisB* leads to loss of cleistothecia production on medium with low levels of tryptophan or histidine, respectively (*Eckert et al., 2000*; *Busch et al., 2001*). These results indicated that the up-regulation of ribosome constituent/biogenesis might be prepared for more proteins synthesis and the increased demand of proteins during the stage of fruiting body maturation. In FB vs. CM, we found DEGs were enriched in carbohydrate metabolic process and pyridine—containing/organic hydroxy compound biosynthetic process. Carbohydrates are one important constituent of the fungal cell wall, like α-1,3-glucan (*Zonneveld, 1972*). It was proposed that carbohydrates were stored during vegetative growth to be utilized as a carbon source for sexual development (*Janssen et al., 2008*; *Martin et al., 2010*). Some study showed that higher metabolic demand of fruiting body development in *N. crassa* (*Martin et al., 2010*). Hence, in *O. sinensis*, carbohydrates

might be required for mycelium growth, as well as a preparation for the later fruiting process.

PCA analysis showed that the expression profiles at fruiting body development stages (FB and DF) more closely resembled each other than that of CM, consistent with the result of DEGs analysis. Furthermore, hierarchical clustering analyses of all these DEGs across different growth stages showed that genes involved in ribosomal structure/biogenesis, translation, starch/sucrose metabolic process and polysaccharide catabolic process were at the highest expression level in FB compared to DF and CM, indicating that more new protein synthesis might be needed for fruiting body maturation. On the other hand, starch and sucrose metabolism are associated with the production of energy and active compounds, such as sugars and volatile secondary metabolites in fruiting body (*Janssen et al., 2008*). Six genes involved in 'starch metabolic process' were at significant and the highest expression level in FB, including glycogen synthase gene (*MSTRG.2946*), a putative glycoside hydrolase gene (*MSTRG.3495*), and one gene encoding glucoamylase I precursor (*MSTRG.5107*), etc. Polysaccharide, an active metabolite, has been reported to be important growth factors in fungi, like *Tuber melanosporum* (*Tisserant et al., 2011*) and *Trichoderma reesei* (*Martinez et al., 2008*). Polysaccharides from the fruiting body of *Cordyceps militaris* have been proven to have antioxidant—and immune—stimulating activities (*Yu et al., 2007*; *Lee & Hong, 2011*). These results revealed more energy consumption and secondary metabolites production in FB compared to DF and CM. Besides, genes involved in ATP catabolic process were found to be at much higher expression level in the stages of fruiting body formation (DF and FB) compared to CM. Previous studies showed that ATP-dependent helicase mutant of *Cordyceps militaris* showed poor production of fruiting body (*Zheng, Qiu & Han, 2015*). And $Ca^{2+}$ signaling pathway ($Ca^{2+}$ transporting ATPase included) was found to be important for fruiting body maturation in *Hypsizygus marmoreus* (*Chen et al., 2018*). Mutants in several subunits of complex 1 of respiration chain composed of proton-pumping NADH ubiquinone oxidoreductase were sterile (*Fecke et al., 1994*), which displayed similar phenotype with respect to the lack of energy. Hence, these results revealed an increased demand of energy and metabolites in fruiting body development.

Alternative splicing contributes to protein diversity. In this study, some basic AS events, including exon skipping (SKIP), intron retention (IR), alternative 3′ splice receptor (A3SS) and alternative 5′ splice donor (A5SS) (*Wang et al., 2015*), were analyzed. IR is the major event in *O. sinensis,* similar to that in *Cordyceps militaris* (*Yin et al., 2012*), while lower than that in *Aspergillus oryzae* (91.56%) (*Wang et al., 2010*) and *Tuber melanosporum* (94%) (*Tisserant et al., 2011*). Moreover, IR-related differential splicing genes (DSGs) are the major class in all types of DSGs, indicating that IR might play an important role in transcriptional regulation during development. Besides, 173 shared DSGs among the comparisons were identified, such as Hsp 90 binding chaperon (*mod-E*, *MSTRG.8149*) and a eukaryotic translation initiation factor gene (*MSTRG.6510*). Mutation of *mod-E* affects the sexual cycle and suppresses vegetative incompatibility in *Podospora anserina* (*Loubradou, Bégueret & Turcq, 1997*). More recently, CG methylation was found to primarily plays a repressive role on expression in two *Pleurotus* mushroom species (*Wen et al., 2019*). So differential splicing genes and transcriptional regulation might

play significant roles in fruiting body development and require further investigation of transcriptional regulatory mechanisms during development in *O. sinensis.*

Fruiting body formation is accompanied by many morphologically distinct cellular differentiations (*Pöggeler, Nowrousian & Kück, 2006*). Some genes for cell cytoskeleton structure and organizations have been proven to contribute to sexual differentiation in *A. nidulans* and *P. anserina* (*Bouhouche et al., 2004*; *Upadhyay & Shaw, 2008*). Our data found that the expression levels of some cell cytoskeleton genes were much higher in both FB and CM than that in DF, such as one actin gene and two tubulin/Fts Z protein gene. Previous studies found that both actin and actin-binding protein fimbrin are critical in the growth of hyphae in *A. nidulans* (*Shestakova, Singer & Condeelis, 2001*; *Upadhyay & Shaw, 2008*). Hence cell cytoskeleton genes might play an important role in vegetative growth and sexual development in *O. sinensis.*

*O. sinensis* exclusively distributes in the harsh alpine environment of the Tibetan Plateau of Asia. Fruiting body development is also regulated by the special ecological stimulus. In general, the mycelia of *O. sinensis* grows in the dark, while fruiting body grows under the light to maturity (*Zhong et al., 2018*), indicating that light is required for fruiting body maturation. Previous studies showed that blue light initiates fruiting body formation in *Coprinopsis cinerea* and *Schizophyllum commune* (*Durand, 1985*; *Purschwitz et al., 2006*). The molecular mechanism underlying light-regulated fruiting body development has been partly elucidated in some basidiomycetes, such as *Aspergillus nidulans* (*Bayram et al., 2010*) and *Neurospora crassa* (*Oda & Hasunuma, 1997*). In this study, several light-regulated genes were also identified, such as nucleotide diphosphate kinase gene (*ndk),* NADPH oxidase gene (*nox*), superoxide dismutase gene (*sod*) and cytochrome oxidase gene (*cox*). Mutants of the NADPH oxidase gene, *noxA,* were found to be sterile in *Aspergillus nidulans* (*Lara-Ortíz, Riveros-Rosas & Aguirre, 2003*). In *P. anserina*, the mutant of *PaNox1*, the ortholog of *noxA,* no longer developed to mature fruiting bodies (*Malagnac et al., 2004*). Both *paNox1* and *paNox2* were also required for controlled production of superoxide and peroxide during sexual development (*Malagnac et al., 2004*). These results demonstrated that Nox catalyzed production of ROS, which is critical for sexual fruiting body development in filamentous fungi. Besides, SOD-1 was required for correct fruiting body morphology in *N. crassa* and it could be due to that SOD-1 is involved in generating a light-dependent ROS gradient that controls neck positioning (*Yoshida & Hasunuma, 2004*). Besides, mutant of cytochrome oxidase (COX) gene showed the delayed perithecial formation and reduced ascus production in *Podospora anserina*, indicating that it is related to sexual development (*Stumpferl, Stephan & Osiewacz, 2004*). Both COX and SOD regulate the level of ROS, suggesting that ROS gradient might participate into fruiting body development. NDK was required for light-dependent neck positioning on the perithecia in *N. crassa* and possibly involved in a light signal transduction pathway (*Ogura et al., 2001*). To summarize, light impacts the fruiting body development in *O. sinensis*, which might be mediated by ROS pathway. Further study is required to investigate the molecular mechanisms underlying ROS–mediated fruiting body development in *O. sinensis*. Besides, some cold–regulated genes were identified, such as fatty acid synthase gene (*cel-2*), acyl-CoA desaturase gene, two potentially encoding heat-shock protein (Hsp) 70 chaperone genes, and four Hsp 101

genes. A previous study showed that *cel-2* mutant of *N. crassa* led to fewer perithecia or rare ascospore (*Goodrich-Tanrikulu et al., 1999*). Mod-E, a Hsp 90 homolog, was reported to be involved in the temperature-dependent sexual development in *P. anserina* (*Loubradou, Bégueret & Turcq, 1997*). However, few related studies were reported (*Pöggeler, Nowrousian & Kück, 2006*). Therefore, it remains to be determined whether *mod-E* or other (heat-shock) proteins are involved in temperature-dependence of fruiting body development and the molecular mechanism.

The fruiting body development was a complex cellular differentiation process, which was regulated by specific environmental stimuli, and subsequently regulated by signal transduction pathways. Mitogen-activated protein kinase (MAPK) modules have been identified in *N. crassa* by genome analysis (*Borkovich et al., 2004*). MAPK module includes three kinases that establish a sequential activation pathway comprising a MAPK kinase kinase (MAPKKK), MAPK kinase (MAPKK), and MAPK (*Kronstad et al., 1998*; *Widmann et al., 1999*), among which three different MAPKs and two different MAPKKKs have been shown to be involved in the fruiting body development in different mycelial ascomycetes (*Pöggeler, Nowrousian & Kück, 2006*). Some MAPKKK cascade protein kinase, like *A. nidulans* SteC, regulate conidiophore development and are required for cleistothecial development (*Wei, Requena & Fischer, 2003*). In plant pathogenic fungus, *Magnaporthe grisea* and *Fusarium graminearum*, the homologs of MAPKKK protein genes, are essential for plant infection and female fertility (*Hou et al., 2002*). In our data four of five MAPKKK genes were up regulated in the stages of fruiting body formation (FB and DF) compared to CM. Two WD domain containing protein (MAPK binding protein) genes were expressed at much higher levels in FB than that in CM. *Cpc 2* encoding WD protein has been proven to control sexual development and amino acid biosynthesis in *N. crassa* (*Müller et al., 1995*). Additionally, genes encoding serine/threonine-protein kinase, which act upstream of MAK-2 during fruiting body development through regulating conidiophore development, are essential for cleistothecial development (*Pandey et al., 2004*). Besides our data found that one cAMP-dependent protein kinase gene was up-regulated in CM compared to DF, suggesting that the PKA signal pathway might play roles in the vegetative growth of *O. sinensis*. However, it needs more data to prove it. To summarize, the fruiting body development in *O. sinensis* might be more dependent on MAPK cascade signaling pathway than on PKA pathway, which is similar to *C. militaris* (*Yin et al., 2012*). Further studies are needed to investigate the MAPK-dependent pathway of fruiting body development in *O. sinensis*.

## CONCLUSION

In this study, Illumina sequencing was performed to investigate the transcriptomes from three serial growth stages of artificially-cultivated *O. sinensis* (CM, DF and FB), which provided large numbers of transcripts to investigate the biology of fruiting body development. The transcriptomic profiles of the stages of fruiting body development (FB and DF) closely resembled compared to the undifferentiated CM stage. Hierarchical clustering analysis and GO enrichment analysis indicated that fruiting body formation

is energy and protein intensive. Also, more various proteins, secondary metabolites and active compound might be produced in mature fruiting body. Furthermore, some ecological factors regulated genes were identified, suggesting that light-regulated fruiting body development might be via ROS-mediated pathway. For the first time, the altered expression levels of some cytoskeleton genes across the three growth stages were identified in this study. In addition, differentially splicing genes might play an important role in fruiting body development of *O. sinensis*. Overall, our work provided a comprehensive overview into fruiting body development of *O. sinensis* and built a foundation for unraveling molecular basis of fruiting body development, which will help the large-scale cultivation of *O. sinensis* in the future.

## Abbreviations

| | |
|---|---|
| **FB** | mature fruiting body of *O. sinensis* |
| **CM** | the mycelium of *O. sinensis* |
| **DF** | developing fruiting body of *O. sinensis* |
| **NGS** | next-generation sequencing |
| **RNAseq** | RNA-sequencing |
| **log$_2$FC** | log$_2$fold-change |
| **DEG** | differentially expressed genes |
| **SEG** | specifically expressed genes |
| **EC** | enzyme codes |
| **PDA** | potato dextrose agar |
| **SRR** | clean reads of *O. sinensis* from the Sequence Read Archive |
| **AS** | Alternative splicing |
| **SKIP** | exon skipping |
| **IR** | retention of single/multiple intron |
| **A3SS/A5SS** | alternative 3′ splice receptor/alternative 5′ splice donor |
| **DSG** | differential splicing genes |
| **DAS** | differential alternative splicing |

### Funding

This study was supported by the Natural Sciences Foundation of China Science (81872959, 81373920, 30801522), the Yong Science and Technology Innovation Team of Sichuan Province (2019CXTD0055), and the China scholarship foundation (201708510027). The funders had no role in study design, data collection and analysis, decision to publish, or preparation of the manuscript.

### Grant Disclosures

The following grant information was disclosed by the authors:
Natural Sciences Foundation of China Science: 81872959, 81373920, 30801522.
Yong Science and Technology Innovation Team of Sichuan Province: 2019CXTD0055.
China scholarship foundation: 201708510027.

## Competing Interests

The authors declare there are no competing interests.

## Author Contributions

- Xinxin Tong performed the experiments, analyzed the data, prepared figures and/or tables, authored or reviewed drafts of the paper, and approved the final draft.
- Han Zhang, Fang Wang and Jing Cao performed the experiments, prepared figures and/or tables, and approved the final draft.
- Zhengyao Xue analyzed the data, authored or reviewed drafts of the paper, and approved the final draft.
- Cheng Peng conceived and designed the experiments, authored or reviewed drafts of the paper, and approved the final draft.
- Jinlin Guo conceived and designed the experiments, prepared figures and/or tables, and approved the final draft.

## Data Availability

Raw Illumina sequencing results of *O. sinensis* are available at the NCBI Sequence Read Archive (SRA): SRR5282569, SRR5282570, SRR5282574, SRR5282575, SRR5282577 and SRR5282578.

The *O. sinensis* strain is available in the China General Microbiological Culture Collection Center, accession number CGMCC 3.14243.

## Supplemental Information

Supplemental information for this article can be found online at http://dx.doi.org/10.7717/peerj.8379#supplemental-information.

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
