# Peer review of "Comparative transcriptome analysis revealed genes involved in the fruiting body development of Ophiocordyceps sinensis"

_PeerJ, doi:10.7717/peerj.8379_

## Round 0.1 · original submission · Major Revisions

We have obtained three very detailed review reports. Please address their comments. Also, pay attention to the annotated PDF provided by reviewer#1 and discuss the portions highlighted therein.

When you prepare your rebuttal, provide the full text of all of the reviewers' comments to the initial version of this submission, interspersed with your detailed replies to each point (preferably in a different font, for ease of reading).

PeerJ requires re-submissions to be accompanied by a copy of the manuscript file with highlighted changes. Please do not highlight those changes manually: use your word-processor built-in "track changes feature" instead, to compare the initial submission to your modified manuscript.

·

Basic reporting

no comment

Experimental design

no comment

Validity of the findings

The transcriptome sequencing and analysis at different developmental stages of the same sample are very important for studying the dynamic development of Ophiocordyceps sinensis. On the basis of transcriptome, some genes related to the development of fruiting bodies were analyzed, which is of guiding significance for future research.

Additional comments

1, As the authors provide, since “Reads of CM, DF and FB were aligned to the O. sinensis CO18 genome using HISAT2”, the authors need to provide matching rate of RNA-seq data on genome, which is helpful for evaluating the quality of sequencing data.

2, Line 90-96, According to the author, RNA data were first compared to the reference genome, and then assembled by stringTie software, resulting in 7518 protein coding genes and 233 putative novel genes. But then, why were 11,497 genes used for comparative analysis with literature data? These data did not match the Table S1. The best way to evaluate the integrity of transcriptome data was to use conservative gene database comparisons, such as BUSCO, which also needed in transcriptome analysis.
And then, now that the author had made a new assembly and produced some new gene, the author should submit the new assembly results to a public database.

3, Line 100-101, Line 166-168 had been described in the materials and methods and can be deleted.

4, In the “2.3. Clustering of gene expression profiles across three growth stages”, The authors should analyze the different gene expression patterns with the phenotypes from hyphae to fruiting body development. Different developmental stages have different physiological functions. Theoretically speaking, such as, the mycelial stage is dominated by vegetative growth and basic physiological metabolism, while the fruiting body stage will have more new protein synthesis, energy consumption, secondary metabolites, morphogenesis, sexual development and so on.

5, Since the reliability of RNA sequencing data were evaluated by qPCR, the results of qPCR and RNA-seq (TPM) should be displayed simultaneously on two coordinate axes to replace Fig5.

6, Line 355-360, What is the difference between DEGs and DETs? In the manuscript, the analysis description of DETs is not presented.

The English quality of the manuscript is not well, especially in the tense problems of different chapters. Usually, the general description of the material method and the result discussion section needs to use the general past tense. It is suggested that the author look for the native English colleagues to polish the manuscript.
What follows is some obvious inappropriateness:
Line 38, “spp.” need not be italicized.
Line 56, “Neurospora. crassa, Aspergillus. Nidulans,” should be “Neurospora crassa, Aspergillus nidulans,”.
Line 66-68 should be given corresponding references.
Line 74, “reveals” should be “revealed”
Line 104 “There are 19” should be “There were 19”
Line 135 “There are seven” should be “There were seven”
Line 278 “2.8. Validation oftTranscriptome” should be “2.8. Validation of transcriptome”
Line 345-346, Is reference 45 referred correctly?

·

Basic reporting

See my recommendations to authors.

Experimental design

See my recommendations to authors.

Validity of the findings

See my recommendations to authors.

Additional comments

Tong et al generated and analyzed RNA-Seq data for Ophiocordyceps sinensis, a medicinal ascomycete with poorly known development. The data generated are useful and will certainly be of interest to people working on fruiting body development in general or on Ophiocordyceps. However, the paper needs a very thorough revision. I’m offering general and more specific suggestions below.

General comments
- Please present read mapping statistics - what % of reads mapped to the reference genome? Also, please clarify if the genome strain or a different strain was used for RNA-Seq. I suspect the latter, in which case, read mapping % could be low due to the divergence of the used strain from the reference genome’s strain.

- I have reservations about the appropriateness of ASProfile method for fungi. This was developed for gene-poor animal genomes and may not perform well in fungi. Fungi need dedicated (e.g. region-restricted probabilistic modeling) pipelines for splicing reconstructions. The authors should examine whether the surprising results they got might have been caused by this.

- The discussion aspect of the paper could be improved. The authors make some interesting observations, but these are hardly placed in a wider or even Ascomycota-specific context. They could take advantage of the close relationship of Ophiocordyceps to Sordaria, Neurospora and Podospora - species with abundant information on fruiting body development.

- The co-expression analyses are neither described in the method nor seem to be correctly done. It seems the analysis had the mature fruiting body stage before the younger stage in the time series (although I should say I find the nomenclature of developmental stages confusing, so I might be wrong here).

- The english should be improved. This is too often said, but the quality of the english sometimes interferes with understanding.

- The conclusions section is hardly a conclusion, rather, its merely a bullet-point listing of main findings. In its current form it does not add any context or synthesis. This is true to the paper in general - it lacks depth, short of exciting discussions or informed speculations.

Specific comments:
l38 - protruding of —> development of
l40 - cleisthothecium —> perithecium
l43 - TCM contrasts with ‘some asian countries’
l49 - ‘favorable’
l49 - costs —> takes
l52 - replace ease by ‘reduce’
l58 - maybe worth mentioning that fruiting body development in Cordyceps species has also been studied extensively, and that fruiting bodies represent a state of complex multicellularity that evolved by unique mechanisms in fungi (see recent reviews).
l66-70 - a better summary of the findings from previous fruiting body transcriptomes would be useful. Why is RNA-Seq from the same stages as before warranted?
l85 - just ‘during development, …’
l87 - delete ‘were submitted…’ because its redundant
l88 - please supply a table that shows read mapping statistics for each sample (# of reads, # of reads after QC, % of mapped reads). % of mapped reads is particularly crucial in this case, if RNA-Seq was not done on the strain for which the genome was sequenced.
l95 - where do the 11497 genes come from? Above you state 7518 genes were detected.
l98 - ‘genes’
l101 - no need to repeat the definition of DEGs here - this is stated in the methods.
l107 - ‘ribosomal structure protein’ is an imprecise term, please specify what this gene exactly is.
l109 - express —> expression …..upregulated —> higher
l111 - no need to state the year of the study.
l112 - this conclusion is too general. there are >200 cyp450s in fungal genomes, it is unlikely that all of them are associated with vegetative growth. Please remove.
l114 - why p<0.005 here? It was p<0.001 above.
l118 - Figure 2: label for the horizontal axis missing. Given the lack of this information, my guess is that the figure shows the counts of genes with the given GO term. This is not enrichment per se, just the counts, so if this is the case, then the figure is not too informative. Consider plotting p-values instead.
l124 - if I understand right, the GO terms mentioned on lines120-123 belonged to genes upregulated in fruiting bodies vs mycelium. If so, why do you conclude these processes are more active in the vegetative mycelium? The opposite would make sense to me. Carbohydrate metabolism could also refer to cell wall modofication and/or synthesys processes.
l130 - a major question for me in all co-expression analyses is why DF follows FB? FB is the mature fruiting body, whereas DF is the developing fruiting body (confusingly called ‘developed’ throughout, would be better to call it young fruiting body).
l135 - no mention of gene co-expression in the methods, so the reader doesnt know how these co-expression modules were inferred. Please specify.
l143 - sentence sounds like an enrichment analysis is in the background, but no test is specified. Please state what test the p-value is coming from.
l146 - delete ‘patterns’ (expression patterns can’t increase)
l152 - starch is also a reserve material for many species, which gets mobilized upon fruiting body growth. E.g. there is extensive literature on starch metabolism in the fruiting bodies of Agaricus bisporus and Coprinopsis. This should be discussed. Also worth discussion is the upregulation of ribosomal constituents - this could suggest increased demand for proteins associated with fruiting body growth.
l156 - this is a stretch and not supported by the data. Please remove.
l159 what is ‘sexual maturation’? Please rephrase.
l162 - cluster7 is the most interesting to me - constantly higher expression in both fruiting body stages than in mycelium. I think the authors could devote more attention to this cluster. E.g. what previously described fruiting body genes are in this cluster? Any Neurospora/Podospora orthologs in this cluster that are involved in sexual development?
l166 - this might not be true for fungi, where the dominance of intron retention events suggests posttranscriptional regulation via the nonsense mediated decay pathway.
l169-174 - the contradiction between alternative splice sites vs intron retention being the most numerous AS event vs most abundant type of affected gene bothers me. I don’t have an explanation for this, but the results is suspiciously surprising. Intron retention is unversally the most common AS event in fungi. I’m wondering if this result was caused by the software used, which is developed for gene-poor animal systems and not for fungi.
l178 - specify the test used for calculating p-value
l226 - ‘anserina’ is consitently misspelled as ‘anserine’
l229 - ‘reveal’ is an overstatement here. Please rephrase.
l238 - ‘homology’ —> homolog
l242 - cite those few studies
l255 - rephrase sentence (yeast ste11 can’t regulate conidiophore development)
l268 - what is relatively? If expression difference wasn’t significant please state so.
l288 - the conclusions section is a mere listing of main findings, there is little discussion or synthesis here.


l307 - please define developmental stages. What is considered developing and mature fruiting body?
l311 - ‘ascus forming’ —> without asci. Also, if you refer to young fruiting bodies, then ‘developing fruiting body’ as the name of the stage would be better.
l312 - perithecium —> perithecia
l314 - please state how many biological replicates were analyzed.
l338 - these perl-scripte might need to be provided as supplemental material by journal policy.
l341-347 - this entire paragraph is not clear. What is the aim here?
l359 - ‘statistical enrichment’ makes little sense here. please provide details on how hierarchical clustering was done.
l377 - ‘was —> ‘were’

Reviewer 3 ·

Basic reporting

No comment.

Experimental design

No comment.

Validity of the findings

Solid.

Additional comments

The manuscript by Tong et al. investigated transcriptomic dynamic changes in Ophiocordyceps sinensis during development of fruiting body formation at three stages. By comparing the expression and splicing profile among different tissues, authors identified a set of genes responsive to the transition of different development stages and validated their expression pattern. The experimental design was solid and the conclusions are well supported by the sequencing data. While their data are of apparent interest to the community, I do have some concerns regarding the results presentation and materials and methods description.
(1) The title of section 2.1 (line 84) is the same as section 2.2 (line 99). May consider to make changes to the title of section 2.1.
(2) After re-assembly of transcriptome data, authors identified 7518 protein coding genes and 233 putative novel genes (line 90), however, the number is 11497 in line 96. Which number is correct?
(3) The software used for the identification of differentially expressed genes (DEGs), should be given. Are they DESeq, EdgeR, EBseq or other softwares? I could find the information any section of the ms.
(4) Statement is improper in line 122. Because asexual mycelium (CM) is the initial stage, may change it to “percentage of down-regulated genes were higher than up-regulated genes”. Please also correct corresponding statements in other places (e.g. line 108, line 274, line 285).
(5) The developmental order is from CM to DF to FB. In line 137, the statement “transition from CM to FB” is improper. Similarly, figure 3B is wrong in development order. Please check the data carefully and make changes accordingly to results of this part (line 135-162).
(6) Statement is not clear in line 173-174. Please define what AS gene is. What does “different” mean here? Does it mean different splicing gene (DSG)? Moreover, how many different alternative splicing (DAS) events related to 937, 946 and 652 DSGs? In figure 4B, the number of IR-related DSG (figure 4B) is similar to corresponding AS event number (figure 4A). Does this mean nearly all AS events are differential spliced? Please recheck the data.
(7) Section 4.2 is ambiguous. Please re-order the analysis steps. For example, lines 343-347 should be before line 339, lines 352-353 should be after line 355, and so on.
(8) Lines 341-343. Please explain why this analysis was done? The results section did not mention this part.
(9) Lines 355-369. The results section do not contain the content related to differentially expressed transcripts (DETs). Please remove it.
(10) Line 369-370. Hierarchical clustering is used for classifying samples.
(11) Line 371-372. What does “Q value” mean? Which method was used? The method of this part is not clear. Also, if differential splicing analysis was done here, please add the corresponding content.
(12) Line 373. What’s the difference between TopGO and the previous method used for GO enrichment analysis? Please use standard method.

Minor:
(1) Lines 55-56. “O.sinensis like fungi” to “Ascomycetes fungi”.
(2) Line 81. change “aids” to a suitable word.
(3) Line 85. “duplicate samples” to “samples with two biological replicates”.
(4) Line 88-89. “more than 63 million paired-end, reads per replicate, were generated with Illumina HiSeq” to “more than 63 million paired-end sequence data from each replicate were generated with Illumina HiSeq 2000 platform”.
(5) Line 89-90. Please cite the original one paper of HISAT and Stringtie.
(6) Line 107. Remove “, p < 0.001” and do it in other places.
(7) Line 115. Please confirm your criteria of DEG and enrichment analysis: p-value or FDR? 0.01 or 0.001 or 0.005? They variated in different places (including supplementary tables).
(8) Line 124. “during” to “in”?
(9) Line 131-132. Which figure or table supports your result?
(10) Line 133-134. Only abbreviation of development stage is enough.
(11) Line 176. “AS genes” to “DSG”.
(12) Line 214-215. Remove “, FDR < 0.001” and do it in other places.
(13) Line 248. “MAPK ” to “MAPK”.
(14) Line 253. Please give exact gene number.
(15) Line 269. “Transcript Factors” to “transcript factors”
(16) Line 280. “FB and CM stages” to “three development stages”.
(17) Line 296. “that” to “than”.
(18) Line 303. “enriches” to “supports”.
(19) Line 340. “scaffold” to “transcripts”.

---

## Round 0.2 · Minor Revisions

There are still a few things to polish in your manuscript. Please address the remaining issues.

·

Basic reporting

no comment

Experimental design

no comment

Validity of the findings

no comment

Additional comments

Please reconfirm the data of Table_1_Mapping_results_of_O._sinensis_transcriptomes and provide detailed and valuable information as much as possible.
One of the two data (Raw Reads and Clean Reads in Table_1.....) seems to be mistake.

Reviewer 3 ·

Basic reporting

Ok.

Experimental design

Ok.

Validity of the findings

Ok.

Additional comments

In general, I supportpublication of thsi manuscript. However, authors only addressed some of the issues I raised for hte previous version. I think these thould be done. I have the following additonal comemnts:
(1) Importantly, authors did not do as I suggested previously reorganizing the order of comparisons among three stages (CM-DF-FB). The order should be: DF vs.CM, then FB vs. DF and the last FB vs.CM in the main text. Please also re-organize the figures based on the right order (such as Figure 1A and 1B, Figure 2, Figure 4 and Table 1).

(2) Please add the method of clustering analysis of gene expression in Materials and Methods section.

(3) Line 151-152. The definition of A5SS and A3SS are wrong. So are in line 430-431.

(4) Statement is still improper in line 202-203. Maybe it could be that “103 and 58 genes were up-regulated and down-regulated, respectively, in FB” because DF is a former stage and FB is a latter stage. Please also correct corresponding statements in all other places (e.g. line 214-215, line 220-221 and so on).

(5) Line 236, “transcripts” or “genes”? (Also, in line 238, 242, 244 and etc.).

(6) Line 271. Please give the full name of different AS events (exon skip, ES; intron retention, IR; alternative donor, A5SS and alternative acceptor A3SS and so on).

(7) In Figure 4, the x-lab of A doesn’t match to that of B and C. Please give the full information of different types of AS events. Or only give the number of AS events among A3SS, A5SS, IR and ES in Figure 4A. Meanwhile, please correct corresponding content in main text (including Materials and Methods and Results).

(8) Legend of Figure 5. FDR < 0.05 or 0.001?

(9) Theerare new papers on the topic, and these shoudl be referred. For exmaple, Wen et al. (2019). Transcriptome changes during major developmental transitions accompanied with little alteration of DNA methylome in two Pleurotus species. Genes 10 : art no. 465

---

## Round 0.3 · Minor Revisions

I noticed that Figure 4, which depicts the number and kind of alternative splicing events, has changed drastically /from up to 20 thousand to less than 3 thousand AS events per condition) between versions0/1 and the current version 2. You have not mentioned this nor discussed the reasons behind it, and this worries me a lot. Please provide a thorough explanation of the reasons for those differences and why you believe the current version is better than the last. Ideally, I would also like you to deposit the input and output of the ASProfile analysis leading to those figures in a public repository (figshare, zenodo, etc.) for full transparency.

Reviewer 3 ·

Basic reporting

Sound.

Experimental design

Sound.

Validity of the findings

Sound.

Additional comments

After a careful check, I consider authors have properly addressed all concerns that I have raised for the precious 2 versions. I have no additonal comments, and would like to endorse its acceptance for publication by the Peer Journal.

---

## Round 0.4 · accepted · Accept

Thank you for your clarifications and for depositing the raw results. I am happy to accept you manuscript for publication in PeerJ!